# Preparation and Characterization of Two Different Liposomal Formulations with Bioactive Natural Extract for Multiple Applications [†]

**Florina Miere (Groza)** [1], **Simona Ioana Vicas** [2,*], **Adrian Vasile Timar** [2], **Mariana Ganea** [3], **Mihaela Zdrinca** [3], **Simona Cavalu** [3], **Luminita Fritea** [3], **Laura Vicas** [3], **Mariana Muresan** [3], **Annamaria Pallag** [3] and **Luciana Dobjanschi** [3]

1   Doctoral School of Biomedical Science, University of Oradea, 1 University Street, 410087 Oradea, Romania; florinamiere@uoradea.ro
2   Faculty of Environmental Protection, University of Oradea, 26 Gen. Magheru Street, 410048 Oradea, Romania; atimar@uoradea.ro
3   Faculty of Medicine and Pharmacy, University of Oradea, 1 Decembrie Street, 410073 Oradea, Romania; mganea@uoradea.ro (M.G.); mzdrinca@uoradea.ro (M.Z.); scavalu@uoradea.ro (S.C.); lfritea@uoradea.ro (L.F.); laura.vicas@gmail.com (L.V.); mmuresan@uoradea.ro (M.M.); annamariapallag@gmail.com (A.P.); dobjanschil@uoradea.ro (L.D.)
*   Correspondence: svicas@uoradea.ro
†   All authors contributed equally to this work.

**Abstract:** Liposomes continue to attract great interest due to their increased bioavailability in the body and because the substances encapsulated are protected while maintaining their effectiveness. The aim of this study is to obtain "giant" liposomes by lipid film hydration using a preparation formula with two different phospholipids, phosphatidylcholine (PC) and phosphatidylserine (PS). Firstly, the macro- and microscopic characterization, total phenols content and antioxidant capacity of the plant *Stellaria media* (L.) Vill. were assessed. Then, *Stellaria media* (L.) Vill. extract was encapsulated in both formulations (PCE and PSE) and the liposomes were characterized according to their morphology, size distribution and Zeta potential using optical microscopy and dynamic light scattering. The encapsulation efficiency (EE%) was determined using the Folin–Ciocalteu method and the values of both formulations were compared. PC and PCE liposomes with a diameter between 712 and 1000 nm and PS and PSE liposomes with a diameter between 58 and 1000 nm were obtained. The values EE% of *Stellaria media* (L.) Vill. extract for PCE and PSE were 92.09% and 84.25%, respectively.

**Keywords:** liposomes; *Stellaria media* (L.) Vill.; film hydration method; phospholipids; Zeta potential; encapsulation efficiency

## 1. Introduction

In recent years, more attention has been paid to bioavailability and pharmacovigilance issues, highlighting the existing inconveniences in the case of conventional pharmaceutical forms. To date, several undesirable aspects have been highlighted regarding the conventional pharmaceutical forms, such as large variations in plasma levels of the drug, the effect of the first hepatic passage, variations in the absorption or aggression of mucous membranes [1].

Thus, in recent years, researchers have sought to obtain new pharmaceutical forms, in order to develop vectors capable of ensuring selective targeting and controlled release of the drug to the target organ or cell (drug targeting) [2]. The vectors used were classified into three major classes according to size and vectoring mechanism: (i) class I vectors: microspheres and microcapsules; (ii) class II vectors: liposomes, nanocapsules, nanospheres and (iii) class III vectors: monoclonal antibodies [3].

Spherical vesicles are liposomes that are formed spontaneously when phospholipids are hydrated and can be unilamellar or multilamellar with various sizes [4]. Liposomes are made up of a lipid (the oily phase that most often contains phospholipids) and a hydrophilic phase that may contain various salts acting as electrically charged vesicles, thus maintaining their stability [5]. These particulate systems represent the future vectors of drugs or substances of therapeutic interest because they have been shown to have many advantages, including high stability in contact with the tissue, releasing targeted and timely encapsulated contents and having a composition that is biocompatible with the human body. Liposomal systems can be additionally loaded with active compounds in order to improve the performance of the encapsulated material in terms of bioavailability, half-life, selective and targeted delivery [6].

Liposomes or nanoliposomes (when below 100 nm in size) are among the first target transport systems that have shown exceptional results for clinical trials and there are many similar systems in the medical field that are currently used and have produced good results [5]. Liposomes with a size between 400 and 2500 nm are considered "giant" liposomes, according to the literature [7].

For example, many pharmaceutical substances (antibiotics, antifungals, anti-inflammatory drugs, etc.) as well as plant extracts (*Callendula officinalis*, *Dracocephalum moldavica*, etc.) have been encapsulated in liposomal systems to date [1]. Furthermore, liposomes have been used as vaccine RNA fraction carriers (COVID-19 vaccine) [8].

In practice, the aim is to obtain these liposomal systems with an inclusion percentage of the desired substance close to 100% and to be physically, chemically and biologically stable [9]. The achievement of these considerations is closely related to factors such as preparation method, composition, applied working technique or strategy and mechanism of bioactive compounds release. Depending on their specific purpose of use, the quality of obtained vesicular systems is expressed by parameters such as size, shape, Zeta potential, inclusion efficiency, storage stability, in vivo stability until release at the target and low toxicity [10].

Generally, these liposomal systems with encapsulated extracts are intended for use as adjuvants to improve health or, in other words, as a less harmful alternative treatment to synthetic drugs [11]. It is necessary to include bioactive plant extracts in liposomes since they should be protected along their pathway to the target organ or cells against certain environmental factors in the human body such as pH and the presence of different enzymes, but also because the solid form of liposomal formulations becomes more stable during storage, and hence, the encapsulated extract is better protected [1,11].

*Stellaria media* (L.) Vill. was chosen as the extract in liposomes due to its rich composition (phenols, flavonoids, sterols, tannins and polysaccharides), which confers a wide range of medical uses, as highlighted in the literature [12,13]. The extract from *Stellaria media* (L.) Vill. can be used both internally and externally. The plant has therapeutic effects such as carminative, antiasthmatic, antihistamine, and antiviral ones (especially in viral hepatitis, etc.) [14]. By external application, the extract can be used to treat dermal diseases such as allergies, dermatitis, burns, wounds, and scabbing (healing effect) [15]. Recently, it has been pointed out that extract of the plant *Stellaria media* (L.) Vill. lowered the level of adipocytes in patients whose obesity or overweight was generated by an increased level of hormones, especially progesterone (the extract was administered orally to mice whose obesity was induced by progesterone administration) [15].

The active compounds from the plant extract can degrade as they pass through the digestive tract due to different pH values. Thus, by including extract from *Stellaria media* (L.) Vill. in liposomes, we want to protect the extract during the storage period and in the case of oral administration ensureing controlled release in the intestine.

The main goal of our paper is to describe the preparation method and formulation of "giant" liposomes using different phospholipids, namely phosphatidylserine and phosphatidylcholine, using the technique of hydrating lipid films (Figure 1). The inclusion of *Stellaria media* (L.) Vill. extract in two different types of liposomes, along with their

characterization in terms of morphology, diameter, surface, electrical charge (Zeta potential) and encapsulation efficiency was also reported. The biological properties of *Stellaria media* (L.) Vill. extract were investigated concerning the total phenols content and the antioxidant capacity using the DPPH (2.2-diphenyl-1-picrylhydrazyl) method.

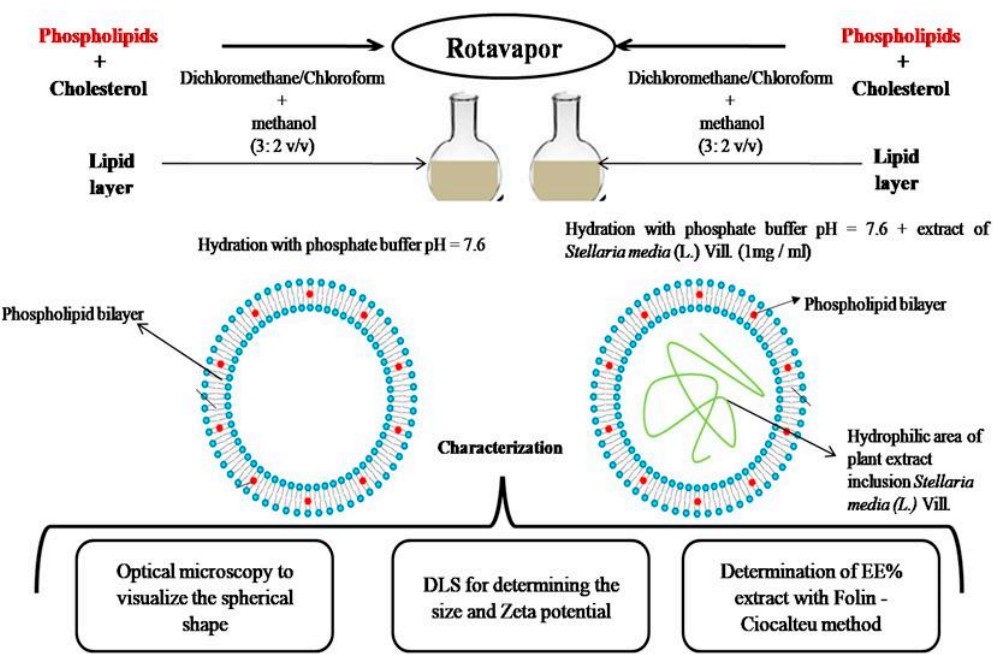

**Figure 1.** The preparation of the liposomes with *Stellaria media* (L.) Vill. extract and empty liposomes by lipid film hydration.

## 2. Materials and Methods

Phosphatidylcholine (Sigma-Aldrich, Saint Louis, MO, USA), phosphatidylserine (Sigma-Aldrich, Saint Louis, MO, USA), cholesterol (Sigma-Aldrich, Saint Louis, MO, USA), phosphate buffer pH = 7.6 (Sigma-Aldrich, Saint Louis, MO, USA), chloroform (Merck, Canada), dichloromethane (Merck, Canada), Triton X 100 (Sigma-Aldrich, Saint Louis, MO, USA), gallic acid (Sigma Life Science, Shanghai, China), 7.5% sodium carbonate (Ingen Laboratory, Timisoara, Romania) and Folin–Ciocâlteu reagent (Merck, Canada) were used as received. Vortex (IKA LabDancer, Merck, Canada), UV-VIS spectrophotometer (Shimadzu MiniUV-VIS, Japan), rotavapor (Heidolph Rotary Evaporator, Laborota, Germany), centrifuge (Hettich EBA200), microscope (Olympus XC30 Attached Microscope, Esslingen, Germany) and a ZetasizerNano ZS (Malvern Instruments, Worcestershire, UK) were the devices used in this study.

### 2.1. Plant Identification, Preparation and Characterization of Stellaria Media (L.) Vill. extract

The plant *Stellaria media* (L.) Vill. was harvested from an unpolluted area in April 2020 in Northwest Romania (Vadu-Crisului region, Bihor County) (at 46° 58′N latitude and 22° 30′ E longitude).

The average maximum temperature in this area is 16 °C and the average minimum temperature is 6 °C.

The identification of the plant *Stellaria media* (L.) Vill. at the time of harvesting was made with the help of a botanical atlas [16]. Subsequently, macro- and microscopic analyzes were performed to highlight the specific elements of this plant. For microscopic analyzes, longitudinal and cross-sections were performed through the strain and were visualized using the OPTIKA B380 optical microscope (Italy).

After identification, the whole plant was harvested, with the total amount of fresh vegetable product being 298.5 g. The plant was dried in an oven at 75 °C until a constant mass was obtained. After drying, 39 g of dried vegetable product was used for the

extraction of the active ingredient with 70% ethanol in a ratio of 1:20 (*w/v*). The mixture was kept for 24 h under continuous stirring in the dark. Then, it was filtered and the solvent was evaporated on a rotary evaporator for 1 h at 44°C and 92 rpm. Subsequently, the remaining aqueous extract was frozen at $-80$ °C [12]. After freezing, the sample was lyophilized and characterized for the total phenol content using the Folin–Ciocalteu method, and the antioxidant capacity was determined using the DPPH method.

### 2.1.1. Total phenols content using the Folin–Ciocalteu method

The total phenols content was determined using the Folin–Ciocalteu method according to the literature [17,18]. Lyophilized extract of the plant *Stellaria media* (L.) Vill. was solubilized in distilled water to form a stock solution (1 mg/mL). A quantity of 0.1 mL of the sample was mixed with 1.7 mL of distilled water, 0.2 mL of Folin–Ciocalteu reagent (1:10 dilution, *v/v*, freshly prepared) and 1 mL of 7.5% $Na_2CO_3$ solution. The mixture was incubated at room temperature in the dark for 2 h. The absorbance was measured at 765 nm using a Shimadzu mini UV-Vis spectrophotometer. The calibration curve was constructed in the concentration range of 0.025–0.5 mg/mL with gallic acid. The total polyphenol content of the extracts was expressed as mg gallic acid equivalents (GAE)/g dry weight (d.w.), using the following equation based on the calibration curve: $y = 2.1913x + 0.0602$ and $R^2 = 0.9999$.

### 2.1.2. The antioxidant capacity of the extract by the DPPH method

The antioxidant capacity of *Stellaria media* (L.) Vill. extract was determined using the DPPH method, as previously described in the literature [17,19,20]. DPPH radical reduction was monitored spectrophotometrically at 517 nm using the Shimadzu mini UV-VIS spectrophotometer.

Briefly, 100 µL of aqueous extract was mixed with 2800 µL of freshly prepared methanolic DPPH solution (80 µM). The samples were kept in the dark for exactly 30 min at room temperature. The percent scavenging of the DPPH radical of the extract was calculated using the following Equation (1):

$$\% \text{ Radical Scavenging Activity (RSA)} = [(A_0 - A_1) / A_0] \times 100 \qquad (1)$$

where $A_0$ was the absorbance of the control and $A_1$ was the absorbance in the presence of the sample (aqueous extract of *Stellaria media* (L.) Vill.).

The experiment was performed in triplicate, and the results are expressed as mean $\pm$ SD (standard deviation).

### 2.2. *Preparation of Liposomes*

Two sets of liposomes were prepared using different phospholipids (Table 1):

- Phosphatidylcholine-based liposomes with (PCE) and without (PC) *Stellaria media* (L.) Vill. extract encapsulated.
- Phosphatidylserine-based liposomes with (PSE) and without (PS) *Stellaria media* (L.) Vill. extract encapsulated.

The lipid film hydration method was applied according to Asprea et al., 2019 [21] with some modifications.

A quantity of 20 mL of organic solvent was added to the lipid mixture used. The lipids were solubilized separately in the mixture of organic solvents and then homogenized. After total homogenization and solubilization of the lipids, the organic solvent was removed by a rotavapor (Heidolph Rotary Evaporator, Laborota 4000) at 37 °C and 103 rotations per minute (rpm) for the PC formulation, and 60 °C and 103 rpm for the PS formulation, taking into account the specific transition temperature (Tc) for each type of phospholipid. Upon total removal of organic solvents, a dry lipid film was obtained, and then hydrated by adding 10 mL of phosphate buffer at pH 7.6. After hydration, the sample was sonicated

for 30 min and then centrifuged for 3 min at 4050 rpm in order to decrease the size of the spontaneously formed liposomes.

The formulation steps for PCE and PSE liposomes were the same as those described above, with the exception that after obtaining the dry lipid film, hydration was achieved using a mixture of *Stellaria media* (L.) Vill. extract and phosphate buffer solution (pH7.6) leading to a concentration of 1 mg/mL of extract.

**Table 1.** Composition formula related to liposomes with and without *Stellaria media* (L.) Vill. extract.

| Composition | PC | PCE | PS | PSE |
|---|---|---|---|---|
| **Phosphatidylcholine: Cholesterol** | 3:1 | 3:1 | - | - |
| **Phosphatidylserine: Cholesterol** | - | - | 3:1 | 3:1 |
| **Phosphate buffer pH = 7.6** | 10 mL | 10 mL | 10 mL | 10 mL |
| *Stellaria media (L.)* **Vill. extract** | - | 1 mg/mL | - | 1 mg/mL |
| **Organic solvents** | Dichloromethane: Methanol (3:2 *v/v*) | Dichloromethane: Methanol (3:2 *v/v*) | Chloroform: Methanol (3:2 *v/v*) | Chloroform: Methanol (3:2 *v/v*) |

PC- phosphatidylcholine liposomes without *Stellaria media* (L.) Vill. extract included, PCE- phosphatidylcholine liposomes with *Stellaria media* (L.) Vill. extract included, PS- phosphatidylserine liposomes without *Stellaria media* (L.) Vill. extract included, PSE- phosphatidylserine liposomes with *Stellaria media* (L.) Vill. extract included.

### 2.3. Optical Observation of the Formulations

Liposomes were observed using an Olympus CX40 inverted light microscope, through a 40× objective in phase-contrast mode, and the images were captured by a Hitachi CCD camera.

### 2.4. Average Diameter and Zeta Potential Measurements by DLS

The dynamic light scattering (DLS) method was applied to determine the diameter, distribution and Zeta potential of the formulated liposomes using a Zetasizer Nano ZS (Malvern Instruments, Worcestershire, UK). Polystyrene cells with an optical path of 1 cm were used for diameter measurements, which were taken in triplicate.

Determination of the surface electric charge or Zeta potential is important because it indicates the stability of the liposomal emulsion [22,23]. The tests were performed for each type of obtained liposome (PC, PS, PCE and PSE) using a disposable folded capillary cell.

### 2.5. Determination of the Encapsulation Efficiency (EE %)

The determination of EE% was performed according to the protocol described by Gibis et al., 2016 [24], with some modifications. Thus, the inclusion efficiency was calculated while taking into account the content of total phenols in the extract, which was determined using the Folin–Ciocâlteu method [17,19], after destroying the liposomal membranes with Triton X-100 0.5% (*v/v*) in order to release extract.

The percentage of EE % was calculated according to the following Formula (2) [24]:

$$EE\ \% = M/M_t \times 100 \tag{2}$$

where M represents the total phenols expressed as gallic acid equivalent (mg GAE/mL) of the *Stellaria media* (L.) Vill. extract encapsulated in the liposomes.

$M_t$ represents total phenols content expressed as gallic acid equivalent (mg GAE/g) extract before inclusion in the liposomal formulation.

## 3. Results

*3.1. Macro- and Microscopic Characterization, Total Phenols Content and Antioxidant Capacity of the Plant Stellaria Media (L.) Vill.*

The macroscopic characteristics of the plant *Stellaria media* (L.) Vill. are presented in Figure 2a,b. According to the analysis of the cross- and longitudinal sections through the stem of the plant, the images in Figure 2c,d were obtained using an optical microscope.

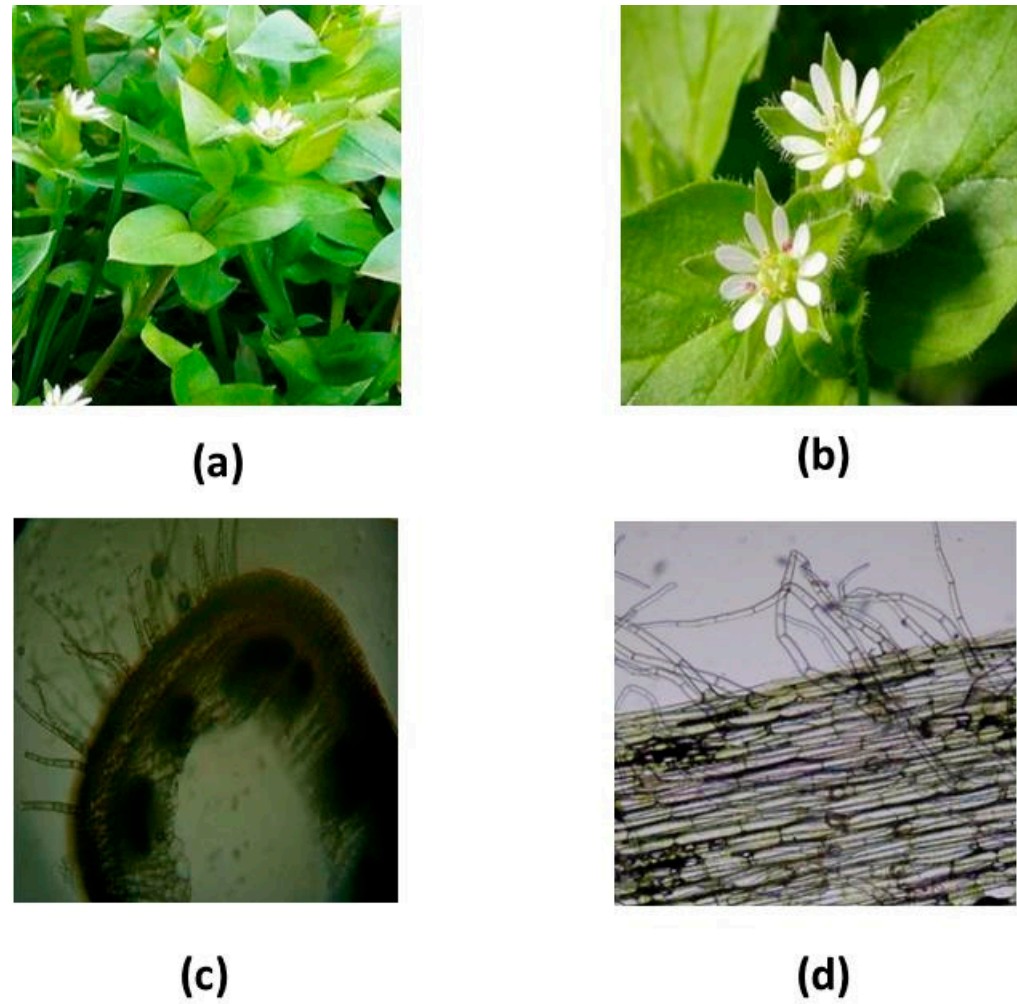

**Figure 2.** Macroscopic and microscopic characteristics of the plant *Stellaria media* (L.) Vill. (**a**) The aerial part of the plant *Stellaria media* (L.) Vill. *Stelariae herba* (personal photo), (**b**) flower of the plant *Stellaria media* (L.) Vill. (personal photo), (**c**) cross-section through the main stem (100×), (**d**) longitudinal section through the main stem (200×)**.**

The total phenols content and % RSA of the plant extract *Stellaria media* (L.) Vill. is presented in Table 2.

**Table 2.** Total phenols of *Stellaria media* (L.) Vill. extract and its antioxidant capacity.

| Total Phenols Content mgGAE/g d.w. | % RSA |
|---|---|
| 17.23 ± 2.31 | 69.19 ± 5.44 |

The total phenols content was determined using the Folin–Ciocalteu method and RSA represented Radical Scavenging Activity.

### 3.2. Liposome Preparation

We have selected two different lipids (phosphatidylcholine and phosphatidylserine) for the preparation of different liposomal formulations. The solubilization was achieved by means of organic solvents (dichloromethane: methanol and chloroform: methanol in a ratio of 3:2 *v/v*).

The liposomal formulations, PC, PS and PCE and PSE, at a ratio of 3:1 phospholipids/cholesterol, was chosen because recent studies showed a better stability was conferred by cholesterol when this ratio was applied, with the role of cholesterol being to stabilize the liposomal membrane [21].

In order to hydrate the lipid film, phosphate buffer at pH 7.6 was used for the two following reasons: the phosphate salts will negatively charge the membrane of the liposomes (thus increasing their stability) and a neutral pH is optimal for spontaneous formation of liposomes [25].

### 3.3. Characterization of Liposomes by Optical Microscopy

Optical microscopy was used to confirm the formation of liposomes using the lipid film hydration method. The existence of spherical vesicles was observed in all the cases. According to the microscopic images, no major differences in shape were observed in the PC formulation compared to PS their homologs, respectively PCE and PSE, as presented in Figure 3.

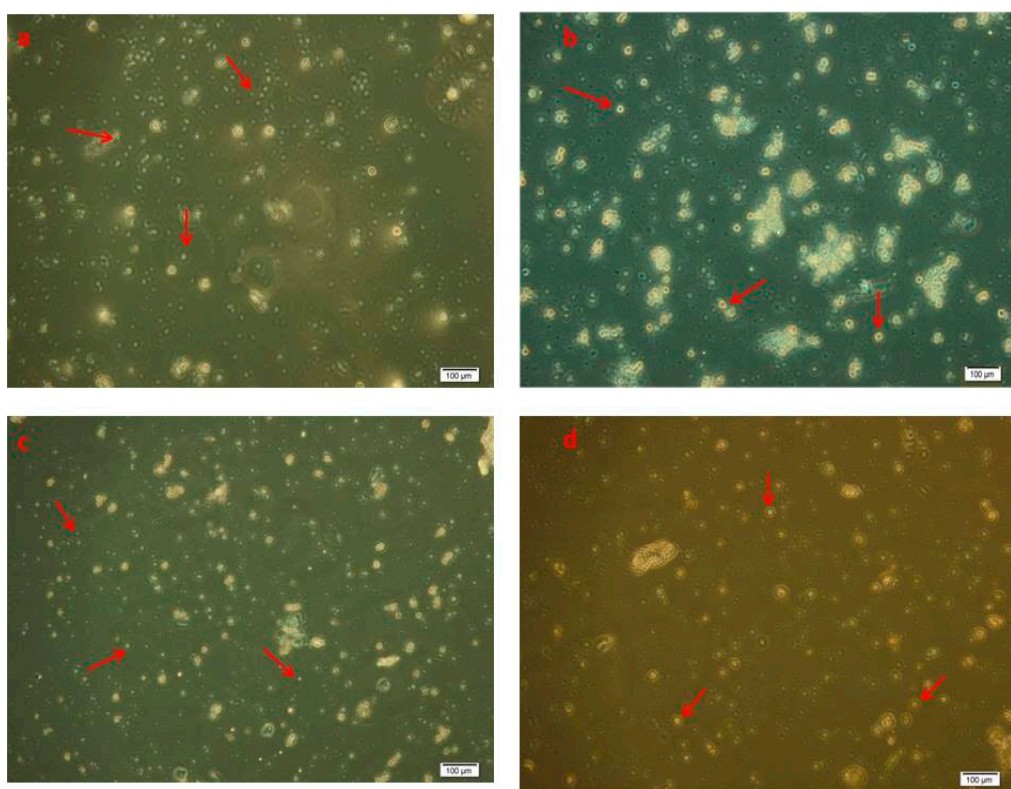

**Figure 3.** Microscopic images of the liposomes: (**a**) Liposomes with phosphatidylcholine (PC), (**b**) Liposomes with phosphatidylcholine with encapsulated *Stellaria media* (L.) Vill. extract (PCE), (**c**) Liposomes with phosphatidylserine (PS), (**d**) Liposomes with phosphatidylserine with encapsulated *Stellaria media* (L.) Vill. extract (PSE). The red arrows show the characteristic spherical shape of liposomes.

### *3.4. Diameter and Zeta Potential Measurements by DLS Analysis*

Microscopy can approximate the diameter of the liposomes, but to determine the diameter interval (or average diameter) of the liposomes, the DLS method was applied and the results are shown in Figure 4 (three repetitions were performed for each sample).

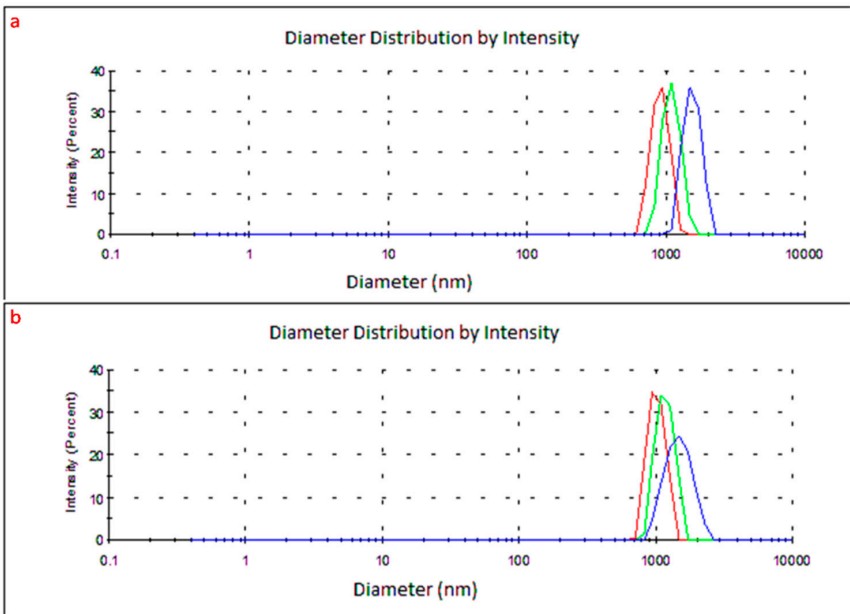

**Figure 4.** Graphical representation of PC liposomes diameter without encapsulated extract of *Stellaria media* (L.) Vill. (**a**) and PCE liposomes with encapsulated extract of *Stellaria media* (L.) Vill. (**b**).

The diameter of PC-type liposomes was between 712 and 1900 nm; 85% of them were in the range 712–1000 nm. The remaining 15% were over 1000 nm, but none were higher than 1900 nm (Figure 5a).

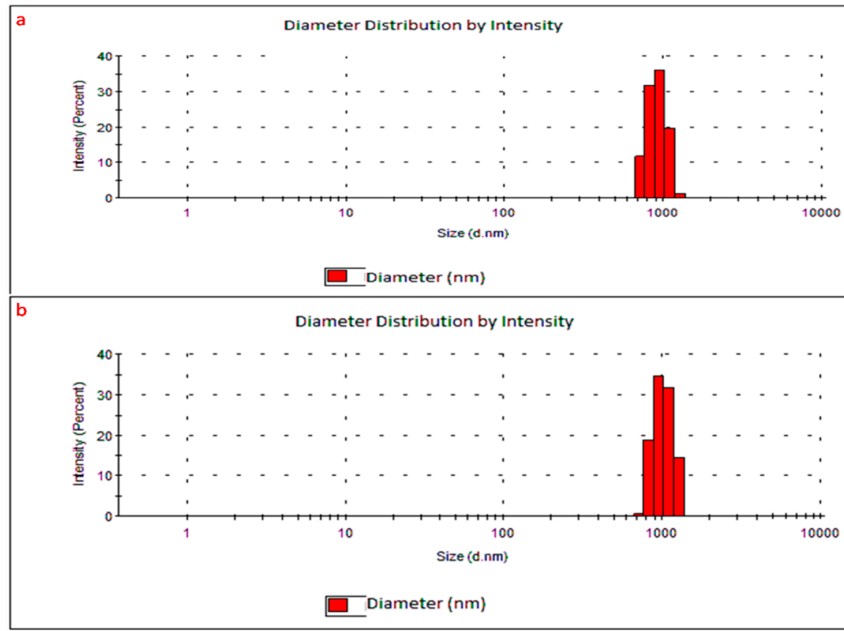

**Figure 5.** Histograms of the diameter distribution for PC liposomes without encapsulated extract of *Stellaria media* (L.) Vill. (**a**) and PCE liposomes with encapsulated extract of *Stellaria media* (L.) Vill. (**b**).

The PCE diameter was also between 712 nm and 2000 nm. A total of 52.5% of the PCE liposomes were between 712 nm and 1000 nm and the remaining 47.5% were over 1000 nm, but none were higher than 2000 nm (Figure 5b).

In the case of PS and PSE liposomes, the diameter was measured by three consecutive repetitions for the same formula, resulting in the values shown in Figure 6.

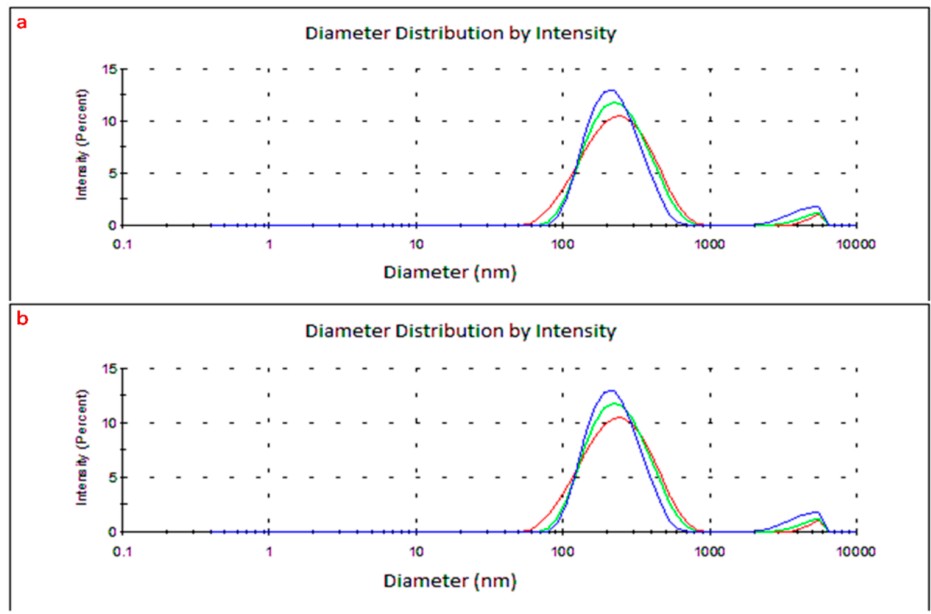

**Figure 6.** Graphical representation of PS liposomes diameter without encapsulated extract of *Stellaria media* (L.) Vill. (**a**) and PSE liposomes with encapsulated extract of *Stellaria media* (L.) Vill. (**b**).

The size of PS-type liposomes was between 58 nm and 2500 nm, with 93% of them being in the range of 58–1000 nm (Figure 7a).

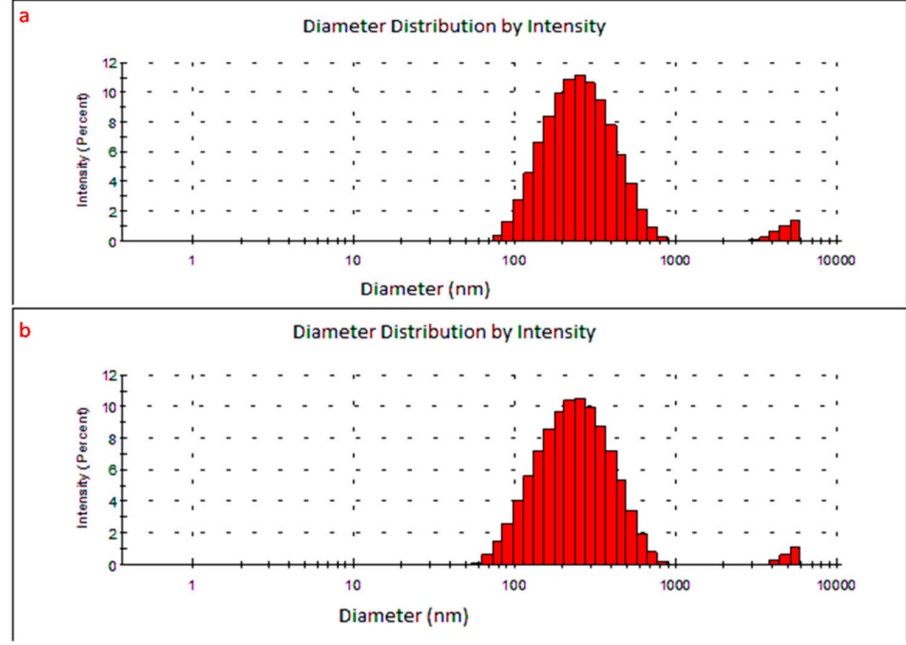

**Figure 7.** Histograms of the diameter distribution for PS liposomes without encapsulated extract of *Stellaria media* (L.) Vill. (**a**) and PSE liposomes with encapsulated extract of *Stellaria media* (L.) Vill. (**b**).

The PSE diameter was between 68 and 2500 nm. A total of 92% of the PSE liposomes had dimensions between 68 and 1000 nm (Figure 7b).

Concerning the surface electric charge (Zeta potential), negative values were obtained for the PS (−2.96 mV) and PSE (−1.93 mV) formulations, while for the PC and PCE ones, the values were positive (+2.01 mV and +3.28 mV, respectively).

*3.5. Encapsulation Efficiency of Stellaria Media (L.) Vill. Extract.*

The encapsulation efficiency of *Stellaria media* (L.) Vill. extract in liposomes (PCE and PSE formulations) was evaluated and a 92.09% inclusion of *Stellaria media* (L.) Vill. extract was obtained for PCE liposomes compared to 84.25% in the case of PSE liposomes.

## 4. Discussion

Figure 2 shows the results of the macroscopic analysis, highlighting the main characteristics of the plant *Stellaria media* (L.) Vill. in the family *Caryophyllaceae*, genus *Stellaria*. The aerial part of the plant and the characteristics of the family it belongs to can be seen in Figure 2a—the leaves are sessile having an opposite orientation on the stem and the stem is articulated. In Figure 2b, the characteristics of the flower can be observed: five petals oriented in the shape of a star (hence the name *Stellaria*) and five sepals covered by protector hairs.

Macroscopic analysis from Figure 2a,b demonstrated that the plant *Stellaria media* (L.) Vill. belongs to the family *Caryophyllaceae*. The genus *Stellaria* includes several plants (*Stellaria alsine*, *Stellaria graminea*, *Stellaria nemorum*, *Stellaria pallida*, *Stellaria langifolia*, etc.) and the plant *Stellaria media* (L.) Vill. differs from the others by the existence of a single series of protector hairs on the stem [26]. Through microscopic analysis, it was possible to demonstrate the existence of these protector hairs and that the harvested plant is indeed *Stellaria media* (L.) Vill. Figure 2c shows the cross-section through the stem and Figure 2d, the longitudinal section of the stem, both highlighting the existing protector hairs.

According to the literature, *Stellaria media* (L.) Vill. is characterized by a highly complex chemical composition consisting of polyphenols, saponosides and vitamin C, which, together, the strong antioxidant activity of this plant is attributed to [26,27]. Phenols rich content and antioxidant capacity of *Stellaria media* (L.) Vill. extract were evaluated in this paper using the Folin–Ciocalteu and DPPH methods.

The bioactive compounds and the antioxidant capacity of *Stellaria media* (L.) Vill. in this study are comparable with the results obtained by the same authors Miere (Groza) et al., 2019 [17] on the same plant, but harvested from another area (around the Crisul Repede river, Oradea city, Bihor county, Romania).

Concerning the liposome formulations, the ratios of phospholipids to cholesterol previously reported in the literature were 1:1 or 1:2 according to Mozafari et al., 2008 [28]. According to Zaka-Ud-Din et al., 1974, there are some advantages of liposomal formulations containing a mixture of phospholipids and cholesterol in different ratios, as the combination was shown to increase the permeability of the liposomal membrane, and consequently, increase the fusion of liposomes with cells in vivo [29].

After hydration, the obtained liposomes were reduced by sonication and centrifugation, as mentioned in the literature [21,30]. According to the literature [7], by hydrating the lipid film, "giant" multilamellar liposomes were obtained, with their major advantages being the increased stability and ease of preparation [7,31].

The images obtained for the PC, PS, PCE and PSE liposomes shown in Figure 3 are comparable to the images found in the literature. For example, Siepmann et al., 2012 and Gibis et al., 2016 aimed to obtain liposomes that were uniformly distributed in solution using the lipid film hydration method and then, visualized their characteristic spherical shape under a light microscope. In all these cases, optical microscopy was used in order to confirm the formation of liposomes [24,32].

Regarding the differences between the liposomes with or without encapsulated substances, it can be observed that their morphology was not significantly changed upon

inclusion, which is consistent with the findings in the literature [28,29]. Therefore, based on the optical microscopy images (Figure 3), it can be stated that the liposomal formulations were successfully obtained and had a stable spherical morphology [33,34].

The diameter is an important parameter as it is responsible for the liposomes' behavior both in vitro and in vivo, as well as their validity and stability [35]. For example, in the literature, it has been shown that "giant" liposomes react better in vivo than those of nanometric size [35]. The explanation for this is that they persist longer in the bloodstream than the nanometric ones, so they can be administrated at a longer time interval [35–38].

It was observed that the size of PCE-type liposomes remained in the same range as that of PC-type liposomes, 712–2000 nm, (Figure 5a,b), which means that the inclusion of *Stellaria media* (L.) Vill. extract in liposomes did not increase their diameter range, nor decrease the percentage of liposomes considered small (58–1000 nm).

It was observed that the diameter of the PSE-type liposomes (Figure 7b) remained in the same range as that of the PS-type liposomes (Figure 7a)—58–2500 nm—which means that the inclusion of *Stellaria media* (L.) Vill. extract in the liposomes did not increase their size range, nor decrease the percentage of liposomes that were considered small (58–1000 nm).

After analyzing the size of liposomes, it can be stated that PS liposomes have a smaller diameter (93% below 1000 nm) than PC liposomes (85% are 1000 nm), but both types of liposomes can be classified according to their size in the category of the "giant" type liposomes [28]. Meanwhile, taking into account the diameter of the PSE (92% below 1000 nm) and PCE liposomes (52.5% below 1000 nm), it can be stated that these liposomal formulations with included extract are also "giant" type.

According to the literature, the molecular weight of the phospholipids used in liposome formulations can influence their size, electrical charge and shape.

Thus, the smaller diameter of the PS- and PSE-type liposomes can be explained as phosphatidylserine is a phospholipid with a lower molecular weight in comparison with phosphatidylcholine, which is characterized by a voluminous molecule [1].

It can be noticed that the molecular structure of the phospholipid also influenced the electrical charge of the liposomes. Negative values of Zeta potential were obtained for PS and PSE liposomes, demonstrating that these types of formulation are more stable than PC and PCE liposomes [1,31].

The EE % of the *Stellaria media* (L.) Vill. extract in the liposomes is comparable with other values concerning encapsulation within liposomes of other plant extracts. The EE% of naringenin was 97.6%, and the EE% of *D.mavavica*, *H. perforatum* and *Callendula officinalis* extracts were 83.98%, 88.3% and 70%, respectively, using the same method for liposome preparation [39].

## 5. Conclusions

In this study, the macro- and microscopic characterization, total phenols content and antioxidant capacity of the plant *Stellaria media* (L.) Vill. were assessed. Two types of liposomes were prepared using the lipid film hydration method with two phospholipids (phosphatidylcholine and phosphatidylserine). Regardless of the formulation, both PC and PS liposomes and their homologues with encapsulated plant extract were "giant" multilamellar liposomes. In the case of PC and PCE liposomes, around 50–80% presented dimensions between 712 and 1000 nm, while more than 90% of PS and PSE liposomes were in the range of 58–1000 nm. The larger diameter of the PC and PCE liposomes confirmed that the type of phospholipids used in the preparation significantly influenced the size and electrical charge of the formulation. The phosphatidylserine-based formulations showed smaller diameters and a negative Zeta potential, meaning they had better stability compared to phosphatidylcholine-based ones. We also demonstrated a high inclusion percentage of the *Stellaria media* (L.) Vill. extract in both formulations—more than 90% for PCE and more than 80% for PSE.

As future perspectives, we propose to study the above-mentioned liposomal formulations PC, PS, PCE and PSE liposomes in terms of their diameter and stability after coating

with $CaCl_2$; the coating of the liposomal membrane with salts or polysaccharides results in better stability and a smaller diameter. Another future approach will be to test the in vitro release, in gastric and intestinal simulated fluids, of the extract encapsulated in the liposomal formulations. Then, we wish to test the in vivo ability of the liposomes with the included extract to reduce the level of adipocytes, leading to a potential use as an adjuvant therapy in diabetes, obesity and overweight (due to hormonal imbalances) or as a dietary supplement for weight loss and weight maintenance.

**Author Contributions:** Conceptualization, methodology and writing of the research were done by F.M., L.F., M.G., A.P., M.Z., and F.M., L.F., A.V.T. and S.I.V. did experimentation and data analysis. S.C., L.D., L.V., and M.M. did supervision, editing and review. All authors have read and agreed to the published version of the manuscript.

**Funding:** This research received no external funding.

**Institutional Review Board Statement:** Not applicable.

**Informed Consent Statement:** Not applicable.

**Data Availability Statement:** Data available in a publicly accessible repository.

**Acknowledgments:** The author (Florina Miere (Groza)) thanks to the University of Oradea, Doctoral School of Biomedical Science, for providing assistance.

**Conflicts of Interest:** The authors declared no potential conflict of interest concerning the research, authorship, and/or publication of this article.

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
