# Peer review of "Preparation and Characterization of Two Different Liposomal Formulations with Bioactive Natural Extract for Multiple Applications†"

_processes, doi:10.3390/pr9030432_

Round 1
Reviewer 1 Report
- I suggest that, in the future, Authors give their text to somebody that did not take part in writing to proof-read it, because some of the spelling and writing mistakes can be avoided in that manner.
- I suggest adding the explanation to why exactly is Stellaria media (l.) Vill. used, and not some other plant?
- I suggest adding future prospects for the using of the encapsulated extracts.
- All of the Images and Tables lack longer description. I suggest the Authors to add more information so that the images could be self-explanatory as they should be.
- There is no scale bar in Figure 2c and Figure 2d - I believe it should be added. I would also suggest that in Figure 2 all of the images be similar in size, as well as the space between them.
- Figure 3 is a bit problematic. I guess that the c) part has moved and therefore it is not the same size as the rest of the images, and there is no scale bar. Also, some of the arrows (in Image 3b and 3d) point to an empty space. I suggest a narrower arrow that can be a bit more precise.
- In the text, the Authors defined liposomes/nanoliposomes as particles with size below 100 nm, but they did not define "giant" liposomes. It can be argued that everything with size higher than 100 nm is "giant" or is there some other classification.
- In order to make the reading clearer, I suggest that the Authors use the term "diameter" rather than "size", it is a bit more precise.

Author Response
Response to Reviewer 1 Comments
The authors would like to thanks for Reviewer 1 comments that improve the manuscript quality.
Point 1: I suggest that, in the future, Authors give their text to somebody that did not take part in writing to proof-read it, because some of the spelling and writing mistakes can be avoided in that manner.
Response 1: The manuscris was checked for type-errors and English reviewed by the authors and an English Teacher.
Point 2: I suggest adding the explanation to why exactly is Stellaria media (l.) Vill. used, and not some other plant?
Response 2: The explanation, why we used Stellaria media (l.) Vill. was included in the Introduction chapter of manuscript in the red colour. Below we have passed the introduced paragraph.
“Stellaria media (L.)Vill. was chosen to be included as extract in liposomes due to its rich composition (phenols, flavonoids, sterols, tannins and polysaccharides)conferring a wide medical use as highlighted in the literature[12, 13]. The extract from Stellaria media (L.)Vill. can be used both internally and externally. The plant has therapeutic effects such as: carminative, antiasthmatic, antihistamine, antiviral (especially in viral hepatitis, etc.) [14]. By external application, the extract can be used to treat dermal diseases such as: allergies, dermatitis, burns, wounds and scab (healing effect) [15]. Also, recently, it has been pointed out that the extract of the plant Stellaria media (L.) Vill. lowered the level of adipocytes in patients whose obesity or overweight was generated by an increased level of hormones, especially progesterone (the extract was administered orally to mice whose obesity was induced by progesterone administration) [15].
The active compounds from the plant extract can suffer degradation along the digestive tract due to different pH values. Thus, by including the extract from Stellaria media (L.)Vill. in liposomes, we want to protect the extract during the storage period, in case of oral administration and for a controlled release in the intestine.”
Point 3: I suggest adding future prospects for the using of the encapsulated extracts.
Response 3: The future prospects for the using of the encapsulated extracts was included in manuscript’s Conclusions:
“As future perspectives, we propose to study the above mentioned liposomal formulations PC, PS, PCE and PSE liposomes in terms of their diameter and stability after coating with CaCl2, the coating of the liposomal membrane with salts or polysaccharides being beneficial for a better stability and a smaller diameter. Another future approach will be to test the in vitro release, in gastric and intestinal simulated fluids, of the extract encapsulated in the liposomal formulations. Then, we wish to test the in vivo ability of the liposomes with the included extract to reduce the level of adipocytes leading to a potential useas an adjuvant treatment in diabetes, obesity and overweight (due to hormonal imbalances) or as a dietary supplement for weight loss and weight maintenance. “
Point 4:All of the Images and Tables lack longer description. I suggest the Authors to add more information so that the images could be self-explanatory as they should be.
Response 4:We made changes (with red color in the manuscript).
Point 5: There is no scale bar in Figure 2c and Figure 2d - I believe it should be added. I would also suggest that in Figure 2 all of the images be similar in size, as well as the space between them.
Response 5: In the Fig. 2c and Fig. 2d was added in the legend of the microscopic images the scale that was worked on. For image 2c it was 100x and for image 2d it was 200x.Our microscope has no possibility to included the scale bar to images.
Point 6: Figure 3 is a bit problematic. I guess that the c) part has moved and therefore it is not the same size as the rest of the images, and there is no scale bar. Also, some of the arrows (in Image 3b and 3d) point to an empty space. I suggest a narrower arrow that can be a bit more precise.
Response 6: The corrections were made!
Point 7: In the text, the Authors defined liposomes/nanoliposomes as particles with size below 100 nm, but they did not define "giant" liposomes. It can be argued that everything with size higher than 100 nm is "giant" or is there some other classification.
Response 7: The argument regarding the “giant” term used for liposomes was included in both the Introduction and Discussions chapters.
The next paragraph was included in Introduction
“The liposomes with a size between 400nm and 2500nm are considered “giant” liposomes, according to the literature [7].”
The next paragraph was included in Discusions:
“After hydration, the obtained liposomes were reduced by sonication and centrifugation as mentioned in the literature[21, 30]. “Giant” liposomes can be unilamellar or multilamellar and this characteristic is given by the method used for formulations, multilamellar “giant” liposomes being obtained by hydrating the lipid film [7].According to the literature [32], by hydrating the lipid film, "giant" multilamellar liposomes were obtained, their major advantages being the increased stability and ease of preparation [7, 31].”
Point 8:In order to make the reading clearer, I suggest that the Authors use the term "diameter" rather than "size", it is a bit more precise.
Response 8: In the Material and Methods, Discussion chapters, and in the Figures 4 - 7 the word `size` was replaced (as suggested) by" diameter ".

Reviewer 2 Report
The publication is a simple copy of the results in literature with no novelity, It shows only that you can reproduce the experiments already described and neither the measurements (zeta, size, DPPH, Folin) nor the procedure have any new aspects. It maybe used as a base (introduction) for a publication where you show new insights in chemical or spherical structure, human relavance or advanced measurements.
Author Response
Response to Reviewer 2 Comments
Point 1:The publication is a simple copy of the results in literature with no novelity, It shows only that you can reproduce the experiments already described and neither the measurements (zeta, size, DPPH, Folin) nor the procedure have any new aspects. It maybe used as a base (introduction) for a publication where you show new insights in chemical or spherical structure, human relavance or advanced measurements.
Response 1:
The manuscripts contains novelty elements because, so far, there are only few research studies about the plant Stellaria media (L.) Vill. We found in the literature very few articles that provide information about this plant from point of view both its phytochemical and biological properties. Furthermore those articles are not very recent [1].Data of literatures included research on different plants of the genus Stellaria (Stellaria palida [2], Stellaria var. lanceolata [3, 4], Stellaria holostea [5], Stellaria nemorum[5], Stellaria yunnanensis [6]), but not on Stellaria media (L.) Vill. [6]. The studies about Stellaria media were also performed on the plant harvested in China [3, 7, 8, 9, 10], Pakistan [11], and we did not find to be studied in Europe (there may be differences).To our best knowledge, there is no microscopy studies performed on this plant. In our article we have shown by the optical microscopy that the plant Stellaria media (L.) Vill. it is the only plant of the genus Stellaria having that series of protector hairs on one side of the stem.
Another novel aspect of this work is the entrapment of the Stellaria media (L.) Vill. extract in liposomes. This extract has never been encapsulated in any lipid vesicles so far (previously the extract was included in alginate beads by the same authors [12]).
So, the novelty of this article consists on the characterization of Stellaria media (L.) Vill. (both the plant and the extract) harvested from a European area concerning the morphological and biological properties and, in addition, the extract inclusion in “giant” liposomes.
- Kitanov, G. Phenolic acids and flavonoids from Stellaria media (L.) Vill. (Caryophyllaceae), Pharmazie 1992, 47, 470-471.
- Verkleij, JAC, de Boer, AM,; Lugtenborg TF. On the ecogenetics of Stellaria media (L.) Vill. and Stellaria pallida (Dum.) pire from abandoned arable field. Oecologia 1980, 46(3), 354-459.
- Peng, Li.; P.; Xia J. Characterization of the complete chloroplast genome of Stellariadichotoma var. lanceolata Bunge, a traditional Chinese medicinal plant. Mitochondrial DNA B Resiur. 2020, 5(4), 3848-38-50.
- Morita, H.; Takeya, K.; Itokawa, H. Cyclic octapeptides from Stellaria dichotoma var. Lanceolata. Phytochemistry 1997, 45(4), 841-845.
- Ancheeva, E.; Daletos,.G.; Muharini, R.; Lin, W.H.; Teslov, L.; Proksch, P. Flavonoids from Stellaria nemorum and Stellaria holostea. Nat. Prod. Commun. 2015, 10(3), 437-440.
- Chandra, S.; Rawat, DS.; Medicinal plants of the family Caryophyllaceae: a review of ethno-medicinal uses and pharmacological properties, Integrative Medicine Research2015, 4, 123-131.
- Zhao, YR.; Zhou, J.; Wang, XK.; Huang, XL.; Wu, HM.; Zou. C. Cyclopeptides from Stellariayunnanensis. Phytochemistry 1995, 40(5), 1453-1456.
- Wang, W.; Su, Z.; Z. Lectotypification of five names in the genus Stellaria (Caryophyllaceae) in China. PhytoKeys 2020, 170, 71-81.
- Su, L.; Jiang, YY.; Liu, B. Oligopeptides in plant medicines cited in Chinese Pharmacopoeia. ZhongguoZhong Yao ZaZhi 2016, 41(16), 2943-2952.
- Rana, D.; Bhatt, A.; Lal, B. Ethnobotanical knowledge among the semi-pastoral Gujjar tribe in the high altitude (Adhwari's) of Churah subdivision, district Chamba, Western Himalaya. J EthnobiolEthnomed 2019, 15(1), 10.
- Shah, NA.; Khan, MR.; Nadhman, A. Antileishmanial, toxicity, and phytochemical evaluation of medicinal plants collected from Pakistan. Biomed.Res.Int. 2014, 384204.
- Miere (Groza), F.; Teusdea, A.C.; Laslo, V.; Fritea, L.; Moldovan, L.;Costea, T.;Uivaroșan, D.;Vicas, S.I.;Pallag, A. Natural Polymeric Beads for Encapsulation of Stellaria mediaExtract with Antioxidant Properties. Mat. Plast.2019, 56 (4), 671-679.
